# Rethinking the Effectiveness of Graph Classification Datasets in Benchmarks for Assessing GNNs

## Abstract

Graph classification benchmarks, vital for assessing and developing graph neural network (GNN) models, have recently been scrutinized, as simple methods like MLPs have demonstrated comparable performance on certain datasets. This leads to an important question: Do these benchmarks effectively distinguish the advancements of GNNs over other methodologies? If so, how do we quantitatively measure this effectiveness? In response, we propose an empirical protocol based on a fair benchmarking framework to investigate the performance discrepancy between simple methods and GNNs. We further propose a novel metric to quantify the effectiveness of a dataset by utilizing the performance gaps and considering dataset complexity. Through extensive testing across 16 real-world datasets, we found our metric to align with existing studies and intuitive assumptions. Finally, to explore the causes behind the low effectiveness, we investigated the relationship between intrinsic graph properties and task labels and developed a novel technique for generating more synthetic datasets that can precisely control these correlations. Our findings shed light on the current understanding of benchmark datasets, and our new platform[1] backed by an effectiveness validation protocol could fuel the future evolution of graph classification benchmarks.

## 1 Introduction

Graph Neural Networks (GNNs) have exhibited superior performance in various domains, including recommendation system (Wu et al., 2022), molecule property prediction (Wieder et al., 2020), and natural language processing (Wu et al., 2021a), etc. To evaluate GNN models in these tasks, specific datasets are often selected as benchmark datasets. Given this mission, a high-quality benchmark dataset should be capable to differentiate the advancements of diverse models. For example, current available benchmarks, such as OGB (Hu et al., 2020), TUDataset (Morris et al., 2020), etc., serve for various link-wise, node-wise, and graph-wise tasks evaluation, as well as graph classification, aiming to automatically discover the optimal method for given tasks. These datasets and benchmark frameworks have immensely facilitated GNN research.

However, recent studies (Errica et al., 2020; Zhao and Wang, 2019; Hu et al., 2020; Dwivedi et al., 2023; Morris et al., 2020) have shown that GNNs may not consistently surpass other baseline methods on specific graph classification tasks. Some simple baseline methods can achieve performance similar to GNNs, and sometimes even better. For example, in the benchmark of (Errica et al., 2020), the MoleculeFingerprint baseline outperforms significantly the widely used GNN models such as GIN (Xu et al., 2018), GraphSage (Hamilton et al., 2017) on three out of four molecular datasets. Nevertheless, despite current research primarily having made significant achievements in analyzing the theoretical expressive power of GNNs(Xu et al., 2019; Feng et al., 2022; Wang and Zhang, 2022) and training schemes (Duan et al., 2022), the reasons for GNNs' failures from these evidences have not been thoroughly analyzed. Few researchers are paying attention to the issues inherent in the datasets themselves.

Therefore, we have adopted a different perspective: dataset compatibility. Our investigation focuses on whether the datasets themselves are suitable for evaluating the advancements of Graph Neural

---

[1]The source code url is: https://anonymous.4open.science/r/Rethink-GNNBenchmark-C0B5/README.md

Networks (GNNs) compared to other methods. This aspect is critical for a fair assessment of whether a GNN method has truly shown improvement. Studies in neural language processing, like (Xiao et al., 2022), define effectiveness as the performance variance across different methods. However, this definition is not directly applicable to graph classification problems. For example, in binary classification problems versus 10-class classification problems, the absolute values of variance are not directly comparable. Hence, in our paper, we reevaluate the effectiveness of existing datasets and attempt to address the following two questions:

**RQ1: Can commonly used graph classification datasets serve the benchmarking purpose which is to effectively distinguish advancements of GNNs compared with other methods?** To address this question, we propose an empirical protocol (Sec.2.2) to investigate the performance disparity between baseline methods and GNN-based methods in terms of the structures and attributes separately by restricting the information input types, i.e., structural information or attributed information. Specifically, we re-organize 16 real-world datasets (Sec.2.1) from common benchmarks across diverse scales and application domains, and conduct extensive experiments with the proposed protocol to investigate the performance gaps fairly on a well-developed benchmark framework extended from (Errica et al., 2020), with our new improvements: 1) supporting datasets from other benchmarks such as OGB, TUDatasets, and synthetic datasets. 2) supporting the construction and combination of various artificial node features, not limited to the framework proposed by (Cui et al., 2022), helps to investigate the impact of different information inputs on the performance of GNNs.

**RQ2: How to measure the effectiveness of existing graph classification datasets?** To answer this question, we design a novel metric (Sec.3.1) to quantify the effectiveness of diverse classification datasets by normalizing the performance gap into a scale-free quantity, with the consideration of the prediction difficulty of datasets and diversity in the number of class labels. The fairness and efficacy of the metric are justified on 16 datasets. For further exploration of the causes of the low effectiveness of datasets, we investigate the relationships between basic graph properties and class labels, and develop a novel approach (Sec.4.2) for generating controllable synthetic datasets, which enables precise control over the degree of correlations between graph properties and class labels. This allows us to study the effectiveness in a controlled environment, providing deeper insights across varying conditions. Additionally, inspired by (Xiao et al., 2022), we further develop a straightforward yet effective regression method to predict the effectiveness (Sec.4.3) of a given dataset.

Our research makes a crucial contribution to the graph learning community, highlighted by the following key points: **(1)** To the best of our knowledge, this is the first work that thoroughly investigates the effectiveness of datasets in the graph learning domain. We provide a novel protocol for systematically evaluating the performance of Graph Neural Networks (GNNs) against baseline methods across a diverse range of graph classification datasets. **(2)** We introduce a pioneering metric to quantify dataset effectiveness, offering new insights for dataset selection in benchmarking graph learning models. This metric serves as a critical tool for researchers to assess and choose datasets more strategically. **(3)** Our work proposes an effective regression method and a unique approach for constructing synthetic datasets to predict dataset performance. This forward-thinking methodology assists in understanding and enhancing dataset selection and design processes.

## 2 EMPIRICAL STUDIES OF EXISTING GRAPH CLASSIFICATION DATASETS

### 2.1 COLLECTION OF DIVERSE DATASETS

To extensive study the performance gap among datasets, in Table 1, we summarize the basic statistics of 16 real-world graph classification datasets collected from commonly used benchmarks, i.e., Open graph benchmark (**OGB**) (Hu et al., 2020) (denoted by ♡), **TUDataset** (Morris et al., 2020) (denoted by ★), and **GNNBenchmark** (Dwivedi et al., 2023) (denoted by ■). These datasets span three major areas, carefully selected to capture significant structural and label diversities. Both attribute and non-attribute graphs are included in this collection. For attribute graphs, features are categorized as node-based or edge-based, listed under "Features" as: node dimension | edge dimension.

**Bio&Chem.** In the fields of biology and chemistry, the ability to predict molecular properties, such as toxicity or biological activity of proteins plays a pivotal role in drug discovery and development. Datasets such as MUTAG, D&D (Yanardag and Vishwanathan, 2015), PROTEINS, and NCI1 furnish a wealth of information for constructing and training machine learning models in these disciplines.

Similarly, in chemical research, datasets like HIV and ENZYMES are indispensable for decoding the interactions between chemical compounds and their potential impacts on living organisms. The large-scale PPA dataset facilitates an understanding of intricate protein interactions and functions, significantly contributing to advancements in personalized medicine and therapeutic approaches.

**Social science.** In the domain of social science, datasets like IMDB-Binary (IMDB-B), IMDB-Multi (IMDB-M), REDDIT-Binary (REDDIT-B), COLLAB are used to study and understand various aspects of social interactions and behaviors.

**Computer vision (CV).** The MNIST and CIFAR10 have been fundamental in shaping the field of computer vision, offering a wide range of images for tasks like object recognition and classification. These two datasets can verify the positional learning ability of GNNs, as the samples are transformed from images into graphs with the super-pixels and coordinates as the node features that inherently carry the positional information of each node.

Table 1: Summary of datasets with different scales, feature types and classification numbers in our experiments.

| Domain | Dataset | Graphs | Classes | Average nodes | Features |
|--------|---------|--------|---------|---------------|----------|
| Bio&Chem | BACE$^\heartsuit$ | 1513 | 2 | 34.09 | 9|- |
| | Tox21$^\heartsuit$ | 7831 | 2 | 18.57 | 9|3 |
| | HIV$^\heartsuit$ | 41,127 | 2 | 25.5 | 9|3 |
| | PPA$^\heartsuit$ | 158,100 | 37 | 243.4 | -|7 |
| | MUTAG$^\star$ | 188 | 2 | 17.9 | 7|- |
| | NCI1$^\star$ | 4,110 | 2 | 29.8 | 37|- |
| | PROTEINS$^\star$ | 1,113 | 2 | 39.1 | 3|- |
| | AIDS$^\star$ | 2000 | 2 | 15.69 | 38|- |
| | DD$^\star$ | 41,127 | 2 | 25.5 | 9|3 |
| | ENZYMES$^\star$ | 600 | 6 | 32.6 | 3|- |
| Social science | IMDB-B$^\star$ | 1,000 | 2 | 19.77 | - |
| | IMDB-M$^\star$ | 1,500 | 3 | 13 | - |
| | REDDIT-B$^\star$ | 2,000 | 2 | 429.61 | - |
| | COLLAB$^\star$ | 5,000 | 3 | 74.49 | - |
| CV | MNIST$^\blacksquare$ | 55,000 | 10 | 70.6 | 3|- |
| | CIFAR10$^\blacksquare$ | 45,000 | 10 | 117.6 | 5|- |

## 2.2 An empirical protocol for evaluating dataset discriminability

We propose a protocol that can fairly evaluate the ability of a graph classification dataset for discriminating the advancements of GNN-based methods over baselines. An effective strategy is to evaluate the performance gap of GNNs over simple baselines. If the performance of GNNs and simple baselines exhibit similarity, it indicates that the dataset lacks the necessary discriminatory power, thus questioning its suitability as a benchmark. The protocol encompasses three main components: (1) the baselines and GNNs for classification; (2) the evaluation framework; and (3) the performance gaps as determined by the evaluation framework. In the rest of this section, we will delve into these three key components and introduce some notations for further usage.

**Baselines and GNNs for graph classifications.** Motivated by the works in (Cui et al., 2022; Errica et al., 2020), we define two types of baselines regarding the input information types, i.e., *structure-dominated baselines* and *attribute-dominated baselines*. As the degree of contribution from node attributes and structures to the model performance varies significantly across different datasets, by constraining the types of model input, we can obtain a more comprehensive analysis and conclusions. In our experiments, we use shallow MLPs as a structure-dominated baseline by feeding the average graph degrees as graph-level features for each sample. For simplicity, we denote any such MLP-based structure-dominated models as $\mathcal{M}_\mathbf{S}^{\text{Baseline}}$. Two attribute-dominated baselines are used in our experiments for attribute-graph datasets. For molecular datasets, we first encode molecules by the encoder suggested in (Hu et al., 2020), then use MoleculeFingerprint model defined in (Errica et al., 2020) for the classification by feeding the molecular encoding features. For non-attribute graph datasets, we simply compose a pooling layer (mean or sum) and shallow MLPs as a baseline. Similarly, we denote an attribute-dominated baseline as $\mathcal{M}_\mathbf{A}^{\text{Baseline}}$. We make comparisons using two typical types of GNNs, as classified in (Wu et al., 2021b), namely, the spatial type represented by graph isomorphism neural networks (GIN), and the spectral type represented by graph convolution networks (GCN).

**Evaluation Framework.** The framework is built upon the benchmarking framework proposed by (Errica et al., 2020). This framework leverages risk assessment and model selection schemes to provide a fair comparison of GNN models using a k-fold cross-validation procedure for model

assessment. Each validation procedure incorporates a model selection process with varying hyperparameters. We further enhance this basic framework in the following ways:

(1) We expand the dataset splitting schemes to support additional strategies, such as the molecular scaffold splitting scheme and user-defined splitting schemes. These offer meaningful, domain-agnostic splits as opposed to random splits.

(2) Our framework allows for the loading of datasets from various sources, including PyTorch Geometric, Open Graph Benchmark (OGB), as well as user-defined synthetic datasets.

(3) Drawing inspiration from the studies (Cui et al., 2022), we have equipped our framework to accept various compositions of graph-level or node-level statistical features as model input, moving beyond the support for only single node labels or edge labels.

**Assessment of performance gap.** GNNs are superior to other neural network structures on graph data because of their ability to capture structure information. However, performance gaps in previous works fail to distinguish the effects of structure and attribute. To solve this problem, we decouple the performance gap into the *structural gap* and *attributed gap*. *Structural performance gap* is denoted by $\delta_{\mathbf{S}}$. It measures the difference in classification accuracy between a structure-dominated baseline and the best performance achieved by structure-agnostic methods, including graph-kernel based approaches and GNNs with artificial node attributes as input features. (e.g., node degree and random noise). *Attributed performance gap* is denoted by $\delta_{\mathbf{A}}$. It quantifies the accuracy difference between an attribute-dominated simple baseline that varies applications and GNNs that utilize real node or edge attributes as input features. Note that GNNs with real node or edge attributes inevitably involve a part of structural information. Formally, a general performance gap is defined as:

$$\delta_{\mathbf{type}}(D, R, \mathcal{M}^{\text{Graph}}, \mathcal{M}^{\text{Baseline}}) = R(D, \mathcal{M}_{\mathbf{type}}^{\text{Graph}}) - R(D, \mathcal{M}_{\mathbf{type}}^{\text{Baseline}}), \ \mathbf{type} \in \{\mathbf{S}, \mathbf{A}\}, \quad (1)$$

where the $R(D, \mathcal{M})$ is the numerical value of given performance metric $R$, e.g., mean classification accuracy or mean AUC-ROC (Area Under the Receiver Operating Characteristics), obtained by model $\mathcal{M}$ on dataset $D$. We use $\mathcal{M}_{\mathbf{type}}^{\text{Graph}}$ to represent a graph-based approach, e.g., a GNN model such as GIN, GCN or a graph kernel method such as Weisfeiler-Lehman graph kernel (WL-GK) (Shervashidze et al., 2011), Subgraph-Matching kernel (SM-GK) (Kriege and Mutzel, 2012), Shortest-Path kernel (SP-GK) (Borgwardt and Kriegel, 2005) in our experiments. These two performance gaps can indicate the discriminating ability of the dataset and provide insights into it. In particular, a narrow gap with high accuracy from both methods suggests the dataset may be too simple to offer discrimination. Conversely, low accuracy from both methods implies the dataset's information is underutilized, necessitating a more advanced approach. A large gap indicates the dataset's strong discriminative power for these two models.

### 2.3 Observations of performance gaps on 16 real-world datasets

Table 2: Mean test accuracy and variations of different methods in 16 graph classification datasets.

| Dataset | $\mathcal{M}_{\mathbf{A}}^{\text{Baseline}}$ | $\mathcal{M}_{\mathbf{A}}^{\text{GIN}}$ | $\mathcal{M}_{\mathbf{A}}^{\text{GCN}}$ | $\mathcal{M}_{\mathbf{S}}^{\text{Baseline}}$ | $\mathcal{M}_{\mathbf{S}}^{\text{GraphKernel}}$ | $\mathcal{M}_{\mathbf{S}}^{\text{GIN}}$ | $\mathcal{M}_{\mathbf{S}}^{\text{GCN}}$ |
|---|---|---|---|---|---|---|---|
| MUTAG | $83.7 \pm 8.35$ | $\mathbf{84.07} \pm 6.26$ | $70.7 \pm 6.89$ | $79.18 \pm 9.83$ | $86.23 \pm 8.50$ | $\mathbf{86.71} \pm 4.67$ | $82.86 \pm 10.43$ |
| PROTEINS | $\underline{74.24} \pm 3.09$ | $70.97 \pm 3.79$ | $73.28 \pm 3.22$ | $60.95 \pm 0.79$ | $\mathbf{72.50} \pm 2.58$ | $68.24 \pm 4.39$ | $64.29 \pm 2.6$ |
| HIV | $96.58 \pm 0.1$ | $\mathbf{96.86} \pm 0.13$ | $96.69 \pm 0.06$ | $96.49 \pm 0.01$ | $51.00 \pm 0.00$ | $\mathbf{96.74} \pm 0.09$ | $96.49 \pm 0.09$ |
| PPA | $20.12 \pm 0.0$ | $\mathbf{24.05} \pm 0.0$ | $16.08 \pm 0.0$ | $9.28 \pm 0.0$ | - | $64.19 \pm 0.0$ | $\mathbf{66.72} \pm 0.0$ |
| D&D | $\underline{76.12} \pm 2.78$ | $72.22 \pm 3.18$ | $70.49 \pm 2.13$ | $62.29 \pm 2.55$ | $62.39 \pm 1.89$ | $62.73 \pm 2.23$ | $\mathbf{64.03} \pm 2.64$ |
| ENZYMES | $29.67 \pm 5.74$ | $\underline{41.78} \pm 3.92$ | $31.72 \pm 4.54$ | $17.56 \pm 1.93$ | $25.00 \pm 3.33$ | $\mathbf{28.33} \pm 4.24$ | $22.0 \pm 3.48$ |
| NCI1 | $66.76 \pm 1.98$ | $\mathbf{80.54} \pm 1.16$ | $76.85 \pm 2.78$ | $50.58 \pm 1.02$ | $62.50 \pm 1.79$ | $\mathbf{75.55} \pm 1.31$ | $65.03 \pm 2.43$ |
| BACE | $68.64 \pm 4.68$ | $\underline{79.95} \pm 2.9$ | $73.5 \pm 3.48$ | $54.33 \pm 0.17$ | $61.84 \pm 0.00$ | $\mathbf{79.82} \pm 3.18$ | $61.32 \pm 5.07$ |
| AIDS | $\mathbf{99.07} \pm 0.85$ | $95.68 \pm 1.53$ | $90.05 \pm 2.27$ | $89.2 \pm 1.16$ | $\mathbf{99.55} \pm 0.52$ | $95.33 \pm 1.16$ | $86.65 \pm 2.24$ |
| moltox21 | $91.05 \pm 0.0$ | $\underline{91.31} \pm 0.0$ | $90.88 \pm 0.0$ | $90.43 \pm 0.0$ | - | $\mathbf{90.53} \pm 0.0$ | $90.45 \pm 0.0$ |
| IMDB-B | NA | NA | NA | $70.63 \pm 3.57$ | $67.10 \pm 4.76$ | $\mathbf{70.8} \pm 2.81$ | $69.5 \pm 2.94$ |
| IMDB-M | NA | NA | NA | $42.31 \pm 4.54$ | $\mathbf{47.00} \pm 5.84$ | $45.93 \pm 4.19$ | $44.98 \pm 4.78$ |
| REDDIT-B | NA | NA | NA | $58.33 \pm 1.18$ | $73.50 \pm 2.05$ | $\mathbf{89.05} \pm 2.14$ | $86.98 \pm 2.52$ |
| COLLAB | NA | NA | NA | $67.68 \pm 0.94$ | $63.92 \pm 1.63$ | $\mathbf{69.92} \pm 1.09$ | $69.79 \pm 1.11$ |
| MNIST | $24.1 \pm 0.33$ | $\mathbf{78.48} \pm 0.72$ | $54.03 \pm 2.15$ | $9.86 \pm 0.01$ | - | $11.74 \pm 1.49$ | $\mathbf{21.93} \pm 0.35$ |
| CIFAR10 | $25.27 \pm 0.6$ | $\underline{49.87} \pm 0.4$ | $45.75 \pm 0.6$ | $10.0 \pm 0.0$ | - | $11.13 \pm 0.99$ | $\mathbf{13.7} \pm 0.62$ |

**Experimental setup.** Utilizing our proposed framework, we assessed 16 real-world datasets following the standard protocol. Due to the limitation of space, all experimental hyperparameters

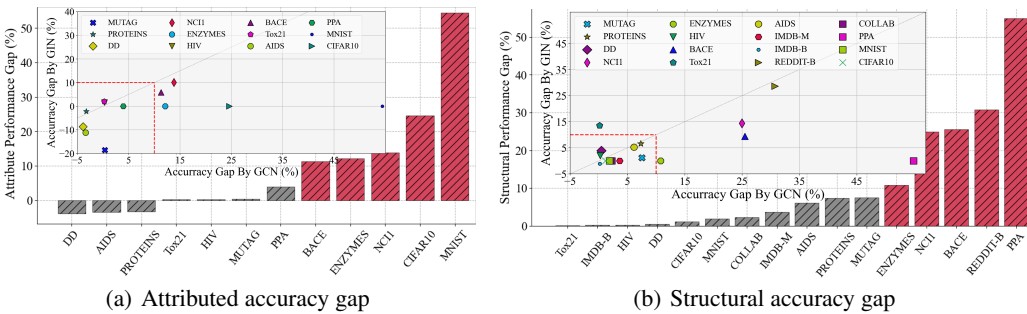

(a) Attributed accuracy gap          (b) Structural accuracy gap

Figure 1: The performance gaps on 16 graph classification datasets are categorized into two types: *Ineffective* (gray) and *Effective* (red) benchmarks. These are sorted in ascending order based on the size of the performance gap. An empirical threshold of 10% is used for categorization, as observed in the inner box of each figure. This box represents the distribution of the accuracy gap for GCN and GIN.

are specified in the Appendix. In Table 2, we show the main experimental results obtained by the protocol over 16 real-world datasets, in which 14 datasets except for PPA and Tox21 were tested by 10-fold cross-validation. The baselines, GNNs and graph kernel methods are introduced in Sec2.2. (Note that, NA denotes the dataset with no attributes, - denotes the dataset is too large to run.) The values that are both bolded and underlined represent the highest accuracy across all attributed and structural models, the solely bolded values indicate the highest accuracy within one type of models. As observed from this table, GNNs excel as the state-of-the-art (SOTA) on the majority of datasets. However, it's important to highlight that the performance gap between the baseline methods and GNNs is minimal for approximately half of the datasets, which is visually represented in Figure 1.

In Figure 1(a), we depict the highest attributed accuracy gap $\delta_{\mathbf{A}}$, comparing the GIN and GCN models. For molecular and protein datasets (HIV, PPA, BACE, and Tox21), we employed a baseline model formed by AtomEncoder (Hu et al., 2020) and MolecularFingerprint (Errica et al., 2020), and solely the MolecularFingerprint model for the other datasets. Subplots provide further comparisons of $\delta_{\mathbf{A}}$ of GIN and GCN across different datasets. Likewise, Figure 1(b) reveals the greatest structural accuracy gap ($\delta_{\mathbf{S}}$) among GIN, GCN, SP-GK, WL-GK, and SM-GK approaches.

From our experimental results, the following observations and insights are derived:

**Observation 1**. Most datasets excel in either attributed or structural performance gaps. Computer vision datasets MNIST and CIFAR10 showcase significant attributed performance gaps, attributable to their dependency on positional and color information of target nodes. Chemical datasets like PPA and Tox21 display noteworthy structural performance gaps due to the inadequacy of average degree information for baseline model predictions, consistent with prior findings (Dwivedi et al., 2023; Cui et al., 2022; Errica et al., 2020; Hu et al., 2020).

**Observation 2**. Datasets displaying huge gaps for both $\delta_{\mathbf{S}}$ and $\delta_{\mathbf{A}}$, like ENZYMES, BACE, and NCI1, reinforce the importance of structures and specific subgraph functions in molecules and compounds. GNNs demonstrate superior performance across most of the datasets by effectively capturing both attributed and structural information simultaneously.

**Observation 3**. Interestingly, among the social science datasets, only REDDIT-B displayed a noteworthy performance gap, indicating a strong correlation between degree information and task labels. This intriguing observation will be further explored and investigated in subsequent sections.

## 2.4 LIMITATIONS OF USING PERFORMANCE GAP AS EFFECTIVENESS MEASUREMENT

In general, half of the 16 graph classification datasets may not be effective to discriminate baselines and GNNs, by using the absolute value of performance gaps. It is important to note that solely relying on the absolute performance gap as an indicator to assess dataset effectiveness could potentially lead to some unfairness and overlook certain inherent limitations.

For instance, two binary classification datasets $D_1$ and $D_2$ have the same performance gap such as 10%, while for $D_1$, the $R(D_1, \mathcal{M}^{\text{Baseline}}) = 80\%$, $R(D_1, \mathcal{M}^{\text{GNN}}) = 90\%$, and for $D_2$, the $R(D_2, \mathcal{M}^{\text{Baseline}}) = 50\%$, $R(D_2, \mathcal{M}^{\text{GNN}}) = 60\%$. It is obvious that the $D_2$ has more complex characteristics leading to the failures of both methods. In that case, we prefer that $D_2$ has more potential performance improvements by using advanced methods, such that it has larger effectiveness. Another limitation is the lack of consideration of the number of class labels. Suppose that $D_1$ has 2 labels, and the $D_2$ has 10 labels, the complexity of datasets is different even when the $R(D_1, \mathcal{M}^{\text{Baseline}}) = R(D_2, \mathcal{M}^{\text{Baseline}}) = 80\%$, and $R(D_1, \mathcal{M}^{\text{GNN}}) = R(D_2, \mathcal{M}^{\text{GNN}}) = 90\%$.

Therefore, in the following section, we will deliberate on the determination of a dataset's suitability for benchmarking purposes and introduce a novel, unified metric designed to measure this degree. This metric takes into account not only the inherent complexity of the dataset and number of class labels, but also the absolute performance gaps observed among different approaches.

## 3 QUANTIFY EFFECTIVENESS OF BENCHMARK DATASETS

### 3.1 EFFECTIVENESS DEFINITION

We define the effectiveness metric $\mathcal{E}$ to quantify the degree of the classification dataset $D$ for discriminating the ability of methods $\mathcal{M}^1$ and $\mathcal{M}^2$ as follows:

$$\mathcal{E}(D) = \sum_{\textbf{type} \in \{\textbf{S},\textbf{A}\}} \frac{|\delta_{\textbf{type}}(D)|}{R^*(|Y| - 1)} \cdot \frac{1 - R^*}{1 - |Y|^{-1}}, \text{ where } R^* = \min(R1, R2). \tag{2}$$

The $R^*$ is the minimal value of two accuracy values from $\mathcal{M}^1$ and $\mathcal{M}^2$, denoted by $R1$ and $R2$ respectively. This formula aggregates two types of effectiveness, each type of effectiveness is the product of two components. The first component is the absolute changing proportion of the performance gap which is normalized by the total number of class labels $|Y| - 1$. This component varies from 0 to 1, if the worst performance is not less than random guessing.

The component $\frac{1 - R^*}{1 - |Y|^{-1}}$, termed the complexity factor and denoted as $\lambda$, ranges between 0 and 1, indicating a dataset's relative complexity. The denominator, $|Y|^{-1}$, reflects random guessing accuracy, with $|Y|$ being the total task labels. The numerator represents the gap between the worst method and perfect classification. If the worst method's accuracy is near $|Y|^{-1}$, $\lambda$ nears 1, indicating high complexity. If it's near 100%, $\lambda$ is close to 0, suggesting a trivial dataset.

Note that, for binary datasets, $R$ can be AUC-RUC or accuracy. This is because the AUC-ROC value for random guessing is 0.5, aligning with $1 - |Y|^{-1}$ when $|Y| = 2$.

### 3.2 PROPERTIES OF COMPLEXITY FACTOR AND EFFECTIVENESS

In this section, we delve into the properties of the complexity factor $\lambda$ and how it manages dataset intricacy considering task label counts. Figure 2 elucidates properties of $\lambda$ and effectiveness $\mathcal{E}$ via variations in Eq. 2.

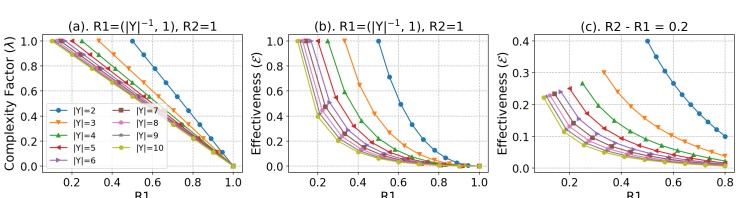

Figure 2: Properties illustration of $\lambda$ and $\mathcal{E}$.

**Property 1:** As the worst method accuracy rises, $\lambda$ linearly decreases (Figure 2(a)). Each curve represents a dataset with task labels from 2 to 10. Essentially, a higher worst method accuracy means a simpler dataset.

**Property 2:** A smaller performance gap leads to a reduced $\mathcal{E}$ (Figure 2(b)). As the gap decreases, the dataset's distinguishability diminishes.

**Property 3:** With a constant performance gap, $\mathcal{E}$ varies based on the worst performance values of $R1$ and $R2$ (Figure 2(c)). Higher accuracies yield a lower $\mathcal{E}$ than lower accuracies. For instance, a 20% difference in accuracy between two methods results in a higher $\mathcal{E}$ if the accuracies are lower.

**Property 4:** For datasets $D_1$ and $D_2$ with the same performance gaps and accuracy, if $|Y_1| < |Y_2|$, then $\mathcal{E}(D_1) > \mathcal{E}(D_2)$ (Figure 2(a-c)). A dataset with more classes has a larger $\mathcal{E}$.

## 3.3 EFFECTIVENESS OF REAL-WORLD DATASETS

We examined the effectiveness $\mathcal{E}$ of 16 real-world datasets using our protocol. Figure 3(a) shows the attributed effectiveness $\mathcal{E}\mathbf{A}$ (in grey) and structural effectiveness $\mathcal{E}\mathbf{S}$ (in red) for all datasets. In Figure 3(b), we assess $\mathcal{E}$ for binary datasets using the AUC-ROC metric. While $\mathcal{E}$ values are consistent across metrics for most datasets, HIV's $\mathcal{E}$ jumps from near 0 to 0.4 with AUC-ROC, emphasizing its

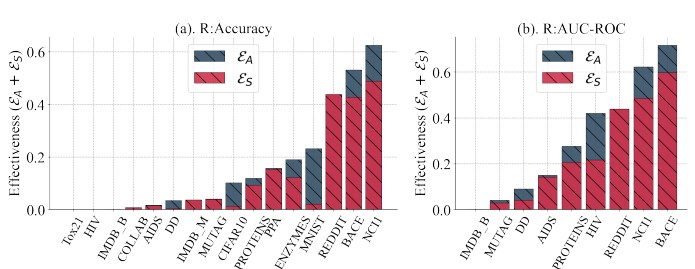

Figure 3: Effectiveness using Accuracy metric and AUC-ROC metric in terms of structural type and attributed type.

suitability for evaluation. Generally, $\mathcal{E}$ remains stable across different metrics. The ranking by effectiveness aligns with the performance gap, confirming a high Spearman correlation between datasets' $\mathcal{E}$ sequences and their performance gap sequences.

In conclusion, by leveraging the definition of $\mathcal{E}$ across various metrics $R$ and models $M$, we can gain valuable insights. These insights aid in the assessment of a dataset's fairness and suitability for benchmarking purposes. Furthermore, this definition guides the selection of appropriate metrics and models for a given dataset, such as opting for accuracy or AUC-ROC as a metric.

## 4 INVESTIGATION OF CAUSES OF LOW EFFECTIVENESS OF DATASETS

In this section, our primary focus is to unravel the reasons behind the low effectiveness $\mathcal{E}$, observed in certain datasets. Then we validate our findings using the controlled synthetic datasets, which allow for systematic evaluation under varying conditions.

## 4.1 CORRELATION BETWEEN GRAPH PROPERTIES AND CLASS LABELS

Inspired by (Cui et al., 2022; Errica et al., 2020; Hu et al., 2020), we hypothesize that for some simple graph properties, they are highly correlated with class labels. This high correlation is what allows simple methods to achieve good accuracy. Therefore, we first examine the correlation between some simple graph properties and class labels.

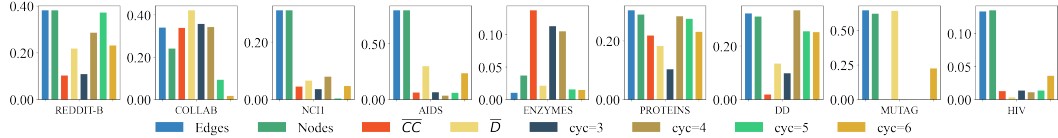

Figure 4: Correlations between graph property sequences and class labels on 9 real-world datasets.

**Graph property sequence.** We generate graph property sequences in terms of some basic graph properties such as number of nodes, average degree, count of cycles, etc. Suppose we have a non-attribute dataset $\mathbb{D}$ with $N$ samples, i.e., $\mathbb{D} = \{g_i\}_{i=1}^N$, and the corresponding labels $\mathbb{Y} = \{y_i\}_{i=1}^n$, where $y_i \in \{0, 1\}$ for a binary classification dataset. Following this sample sequence, we can generate various corresponding graph property sequences. For instance, the average degree sequence, i.e., $\overline{D} = \{\overline{d}_i\}_{n=1}^n$, where $\overline{d}_i$ is the average degree of the graph sample $g_i$. Similarly, we construct the average clustering coefficient (CC) sequence, i.e., $\overline{CC} = \{\overline{cc}_i\}_{i=1}^n$, where $\overline{cc}_i$ is the average clustering coefficient of $g_i$. Besides these two basic properties, we obtain sequences of other different graph properties, i.e., edge count sequence (denoted by Edges), node count sequence (denoted by Nodes), cycle count sequence (denoted by cyc=k), where k represents the cycle length, $k \in \{3, 4, 5, 6\}$.

**Correlation analysis between graph property sequences and label series.** Figure 4 shows the correlations between 8 graph properties and labels $\mathbb{Y}$. The correlation of Edges and Nodes with $\mathbb{Y}$ exceeds 0.2 in most datasets, often above 0.4. In molecular datasets like MUTAG, cycle count is highly correlated with labels, indicating the impact of cyclic structures. Studies (Chen et al., 2020; Rieck et al., 2019; Bouritsas et al., 2022) suggest WL kernels and GNNs struggle to capture substructures, underlining the importance of analyzing graph properties for method performance.

## 4.2 CONTROLLABLE SYNTHETIC DATASETS

Real datasets are finite and insufficiently diverse for an exhaustive exploration of the effects of varying correlations between different graph properties and labels on effectiveness. Existing synthetic datasets (Murphy et al., 2019; Tsitsulin et al., 2022; Chen et al., 2020), present limitations as they rigidly utilize specific properties as labels, unable to adjust the correlation between properties and labels.

We introduce a method to generate controllable datasets, enabling precise modulation of the correlations between any graph properties and class labels. First, we propose a technique to generate random variables with a given correlation coefficient.

**Generate correlated random variables with given coefficients.** Suppose each graph property $\mathcal{P}$, and the class label $\mathcal{Y}$ are random variables, the goal is to sample a graph property sequence $\mathbb{P}$ (e.g., $\overline{CC}$) and the class label sequence $\mathbb{Y}$ from the distributions of $\mathcal{P}$ and $\mathcal{Y}$ respectively, which satisfy a given Pearson correlation coefficient $r$ between the property and label, i.e., $r = \text{Pearson}(\mathbb{P}, \mathbb{Y})$.

**Theorem 1** *Given a set of property variables $\{\mathcal{P}_i\}_{i=1}^K$, each $\mathcal{P}_i$ follows a Gaussian distribution $\mathcal{N}(\mu_k, \sigma_k)$ or Uniform distribution $\mathcal{U}(a_k, b_k)$, and given corresponding Pearson correlation coefficients $\{r_i\}_{i=1}^K$ with label variable $\mathcal{Y}$, with the constraint $\sum_{i=1}^K r_i^2 \leq 1$, then we have:*

$$\mathcal{Y} = \sigma_{\mathcal{Y}} \left( \sum_{i=1}^K n_i r_i + n_0 \sqrt{1 - \sum_{i=1}^K r_i^2} \right), \tag{3}$$

*where $\sigma_{\mathcal{Y}}$ is any desired standard deviation, and each $n_i$ is mutually independent and follows the same distribution as the corresponding $\mathcal{P}_i$ with the same mean value $\mu_i$ but with standard deviation equals to 1. (The proof can be found in Appendix A1.)*

**Inverse graph generation by correlation.** By Theorem 1, we can easily generate a desired dataset (includes some graph properties with specific correlations with class labels) following a dataset construction process (details can be found in Appendix A1 and A2).

The Figure 5 show the precise correlated relationships of generated $\mathbb{Y}$ and each properties $\mathcal{P}_1, \mathcal{P}_2$, and $\mathcal{P}_3$ with different correlation coefficients $r_1 = -0.7, r_2 = 0.1, r_3 = 0.7$ respectively. We demonstrate the different distributions of each property. The properties follow three uniform distributions as shown in the left three boxes, and follow three normal distributions as shown in the right three boxes.

Leveraging Theorem 1, we construct two types of binary classification datasets namely **Syn-Degree** and **Syn-CC**, by generating Erdos–Renyi (ER) graphs and controlling the average graph degree property $\overline{D}$ and average clustering coefficient property $\overline{CC}$ respectively. As shown in Figure 6(a), we construct 9 datasets for each type, where each dataset has 4096 graphs. In Syn-Degree, the $r_i^{\overline{D}}$ and $r_i^{\overline{CC}}$ are both ranging from 0.1 to 0.9. In Syn-CC, all $r_i^{\overline{D}} = 0$, and $r_i^{\overline{CC}}$ ranges from 0.1 to 0.9.

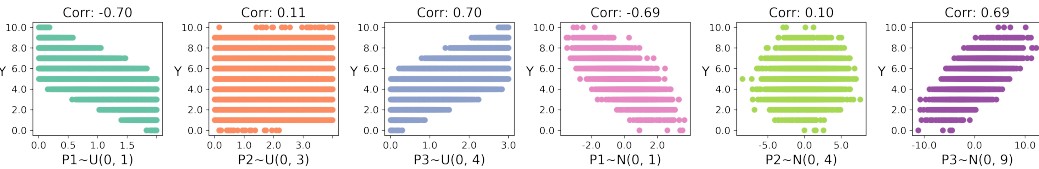

Figure 5: Generated $\mathbb{Y}$ with 11 classes by $\mathcal{P}_2, \mathcal{P}_2, \mathcal{P}_3$ following different Gaussian and uniform distributions with the correlations $r_1 = -0.7, r_2 = 0.1, r_3 = 0.7$ respectively.

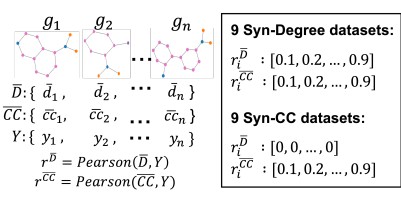
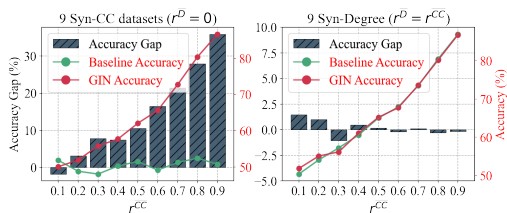

(a) Construction of synthetic datasets    (b) Controllable performance gaps

Figure 6: Controllable performance gaps by two types of synthetic datasets.

Under our framework, the two dataset types showed notable differences in Figure 6(b). As correlation rises, the accuracy gap and GIN's accuracy both increase linearly, with the baseline mirroring random guessing. For the Syn-Degree dataset, GIN's accuracy and the baseline both rise linearly, keeping a minimal gap. This suggests two things: a model's prediction accuracy strongly correlates with the coefficient if it captures a graph attribute linked to the label, and GIN effectively captures clustering coefficient and degree information.

### 4.3 PREDICTING EFFECTIVENESS

Most datasets show strong correlations between graph properties and labels, prompting us to explore predicting dataset effectiveness using these properties, which is computationally cheaper than benchmarking. Drawing from (Xiao

Table 3: Summary of regression results

| Regressor | Real-world datasets | | Synthetic-CC datasets | |
|---|---|---|---|---|
| | Pearson | P-Value | Pearson | P-Value |
| Ridge | $0.80 \pm 0.09$ | $\leq 1e{-}6$ | $0.87 \pm 0.03$ | $\leq 1e{-}6$ |
| SVR | $0.80 \pm 0.09$ | $\leq 1e{-}6$ | $0.89 \pm 0.04$ | $\leq 1e{-}6$ |
| Random Forest | $0.89 \pm 0.03$ | $\leq 1e{-}6$ | $0.87 \pm 0.06$ | $\leq 1e{-}6$ |

et al., 2022), we split each dataset into 10 distinct sets, define 26 features for graph classification, and regress effectiveness using regressors like Random Forest, SVR, and Ridge regression. Using 14 real-world datasets and 9 Syn-CC datasets, we allocate 70% of the splits for training and 30% for testing. Regression performance, verified by the Spearman rank coefficient in Table 3, is based on 10 repeated experiments. Both real-world and Syn-CC datasets show that basic graph properties can effectively predict dataset effectiveness.

## 5 DISCUSSIONS AND CONCLUSIONS

**Limitations and future works.** Our metrics and platform, while innovative, come with certain limitations. The universality of the metrics across diverse datasets and their ability to enable direct cross-comparisons are yet to be fully explored. The utility of the Effectiveness metric based on its magnitude is still an open question, especially given that our comparison is rooted in disparities between only two methodologies. These methodologies, however, span a wide range of learning capabilities, hinting at the potential for a more comprehensive evaluation from multiple diverse methodologies. We also see potential in encouraging graph learning researchers to adopt our Effectiveness definition, broadening the scope with more baselines and methods. Furthermore, our Effectiveness definition and tools could guide researchers in selecting and designing datasets for specific graph representation studies. On a theoretical note, integrating analytical expressions for graph attributes with GNN expressive power might offer deeper insights into GNNs' capabilities beyond the WL-test.

**Conclusions.** Our work provides a detailed analysis of graph classification benchmarks essential for the evaluation and enhancement of GNN models. We introduced an empirical protocol to compare the performance of methods like MLPs to GNNs on certain datasets. Our novel Effectiveness metric serves as a pivotal tool for dataset validation in benchmarking. By devising a method to generate synthetic datasets, we can precisely control the correlation between graph properties and task labels, addressing the issue of low effectiveness in some benchmarks. Our efforts play a significant role in the selection of impactful benchmarks, paving the way for the development of robust GNN models and further advancements in graph learning research.

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

# A  APPENDIX

## A.1  PROOF OF THEOREM 1

We rephrase Theorem 1 here:

**Theorem 1** *Given a set of property variables $\{\mathcal{P}_i\}_{i=1}^K$, each $\mathcal{P}_i$ follows a Gaussian distribution $\mathcal{N}(\mu_i, \sigma_i)$ or Uniform distribution $\mathcal{U}(a_i, b_i)$, and given corresponding Pearson correlation coefficients $\{r_i\}_{i=1}^K$ with label variable $\mathcal{Y}$, with the constraint $\sum_{i=1}^K r_i^2 \leq 1$, then we have:*

$$\mathcal{Y} = \sigma_{\mathcal{Y}} \left( \sum_{i=1}^K n_i r_i + n_0 \sqrt{1 - \sum_{i=1}^K r_i^2} \right), \tag{4}$$

*where $\sigma_{\mathcal{Y}}$ is any desired standard deviation, and each $n_i$ is mutually independent and follows the same distribution as the corresponding $\mathcal{P}_i$ with same mean value $\mu_i$ but with standard deviation equals to 1.*

**Proof:**  The proof is based on Cholesky decomposition of a given covariance matrix. We use two random vectors $\mathbf{X} = (n_1, n_2, ..., n_K, n_0)^T$ and $\mathbf{Y} = (\mathcal{P}_1, \mathcal{P}_2, ..., \mathcal{P}_K, \mathcal{Y})$ to represent the variables in Theorem 1, and suppose there exist a permutation matrix $L$ that satisfy $\mathbf{Y} = L\mathbf{X}$, then we have:

$$cov(\mathbf{Y}) = cov(L\mathbf{X}) = Lcov(\mathbf{X})L^T. \tag{5}$$

Note that, covariance matrix $cov(\mathbf{X}) = \mathbf{I}$, where $\mathbf{I}$ is an Identity matrix, since the standard deviation of $n_i$ is 1 and all $n_i$ are mutually independent. Consequently, the covariance matrix of $\mathbf{Y}$ can be expressed as the product of matrix $L$ and its transpose $L^T$, i.e., $cov(\mathbf{Y}) = LL^T$. Recall the Pearson correlation coefficient of property $\mathcal{P}_i$ and $\mathcal{Y}$:

$$r_i = \frac{cov(\mathcal{P}_i, \mathcal{Y})}{\sigma_i \sigma_y}.$$

Then, we expect $cov(\mathbf{Y})$ to satisfy the given correlation coefficients, i.e.:

$$cov(\mathbf{Y}) = \begin{bmatrix} \sigma_1^2 & 0 & \dots & 0 & r_1\sigma_1\sigma_y \\ 0 & \sigma_2^2 & \dots & 0 & r_2\sigma_2\sigma_y \\ \vdots & \vdots & \ddots & \vdots & \vdots \\ 0 & 0 & \dots & \sigma_K^2 & r_K\sigma_K\sigma_y \\ r_1\sigma_1\sigma_y & r_2\sigma_2\sigma_y & \dots & r_K\sigma_K\sigma_y & \sigma_y^2 \end{bmatrix} = LL^T = \Sigma. \tag{6}$$

To this end, we can use Cholesky decomposition to decompose $\Sigma$ into the product of a lower triangular matrix $L$ and its conjugate transpose $L^T$, since $\Sigma$ is a positive-definite matrix if each $\sigma_i$ is positive. By Cholesky decomposition, we obtain $L$ as follows:

$$L = \begin{bmatrix} \sigma_1 & 0 & \dots & 0 & 0 \\ 0 & \sigma_2 & \dots & 0 & 0 \\ \vdots & \vdots & \ddots & \vdots & \vdots \\ 0 & 0 & \dots & \sigma_K & 0 \\ r_1\sigma_y & r_2\sigma_y & \dots & r_K\sigma_y & 1 - \sqrt{\sum_{i=1}^K r_i} \end{bmatrix} \tag{7}$$

Since $\mathbf{Y} = L\mathbf{X}$, we can obtain $\mathcal{Y} = \sigma_{\mathcal{Y}} \left( \sum_{i=1}^K n_i r_i + n_0 \sqrt{1 - \sum_{i=1}^K r_i^2} \right)$.

## A.2 Algorithm of controllable dataset construction

---

**Algorithm 1:** Controllable dataset construction

---

1 **Input**: $\{r_k\}_{k=1}^{K}$, number of labels $C$,
2    $\{\mathcal{P}_k\}_{k=1}^{K} \sim \mathcal{N}(\mu_k, \sigma_k)$ or $\mathcal{U}(a_k, b_k)$;
3 **Output**:Dataset $\mathbb{D}$ with size $N$;
4 **for** $k = 1$ **to** $K$ **do**
5    Sample $n_k \sim \mathcal{N}(0, \sigma_k)$ or $\mathcal{U}(-\sqrt{3}, \sqrt{3})$ with size $N$;
6    $\mathbb{P}_k = \mu_k + \sigma_k n_k$ or $\frac{a_k+b_k}{2} + \sqrt{\frac{b_k-a_k}{12}} n_k$;
7 **end**
8 Calculate $\mathcal{Y}$ by the Equation 4 ;
9 $\mathbb{Y} = \text{ROUND}(\text{NORM}(\mathcal{Y}) * C)$;
10 $\mathbb{D} = \{(g_i, y_i)\}_{i=1}^{N}$,
11 where each graph $g_i$ has properties $\{\mathbb{P}_k[i]\}_{k=1}^{K}$, and corresponding label $y_i = \mathbb{Y}[i]$;

---

In Algorithm 1, it is easy to prove that $\mathbb{P}_k \sim \mathcal{N}(\mu_k, \sigma_k)$, or $\mathbb{P}_k \sim \mathcal{U}(a_k, b_k)$. The **NORM** function is to normalize the $\mathcal{Y}$ into 0 to 1 by min-max normalization, and the **ROUND** function is to convert $\mathcal{Y}$ from decimal to an integer between 0 and $C-1$, to be used as a class label.

## A.3 More experimental results

### A.3.1 Effects of property correlation on Effectiveness

In Figure 7, we illustrate the relationships between the average clustering coefficient correlations ($\overline{CC}$) and their corresponding Effectiveness measures ($\mathcal{E}_S$) on Syn-CC and Syn-Degree datasets. These relationships are considered under two different metrics: Accuracy and Area Under the Receiver Operating Characteristic Curve (AUC-ROC). We denote the correlation coefficient between the average clustering coefficient sequence ($\overline{CC}$) and the labels as $p^{\overline{CC}}$, and the correlation coefficient between the average degree sequence ($\overline{D}$) and the labels as $p^{\overline{D}}$.

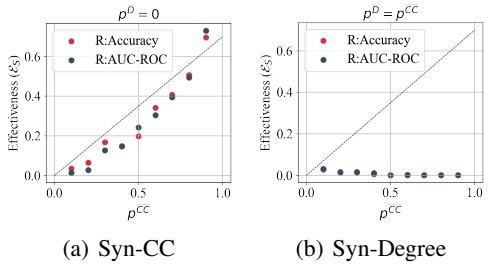

(a) Syn-CC          (b) Syn-Degree

Figure 7: The effects of $\overline{CC}$ correlations on the Effectiveness $\mathcal{E}_S$.

For each dataset in Syn-CC, we set each $p^{\overline{CC}}$ ranging from 0.1 to 0.9, and consistently keep $p^{\overline{D}}$ equal to 0. This results in a strong correlation between Effectiveness and the $p^{\overline{CC}}$, as shown in Figure 7(a). On the other hand, in Figure 7(b), we set $p^{\overline{DD}}$ equal to $p^{\overline{CC}}$, leading to a stable Effectiveness that doesn't vary as $p^{\overline{CC}}$ increases.

These findings indicate a crucial insight: the GIN can accurately learn the average clustering coefficient. This serves as counter-evidence to the example presented in You et al. (2021).

### A.3.2 Effects of different metrics on performance gaps

We examine the influence of different metrics and GNNs on performance gaps, specifically focusing on Accuracy and Area Under the Receiver Operating Characteristic Curve (AUC-ROC), as well as Graph Convolutional Network (GCN) and Graph Isomorphism Network (GIN). The results are presented in Figure 8. Our findings align with those of the Open Graph Benchmark (OGB) study Hu et al. (2020), corroborating that for certain class-imbalanced datasets such as BACE and AIDS, AUC-ROC serves as a more appropriate metric. This is confirmed by the observed large performance gaps. Furthermore, both GIN and GCN exhibit similar performance gaps across most datasets, with GIN notably outperforming GCN in almost all cases.

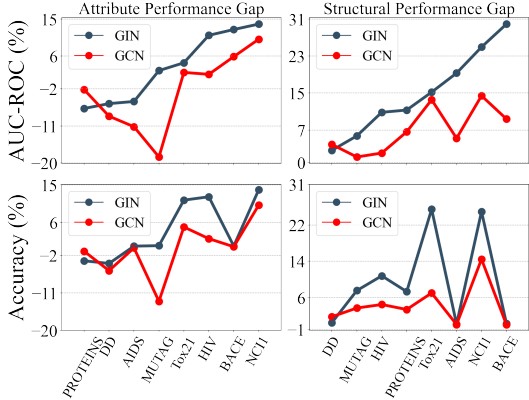

Figure 8: Comparison of performance gaps across two different metrics and GNNs (GCN and GIN).

### A.3.3 VISUALIZATION OF REGRESSION RESULTS OF EFFECTIVENESS

In Figure 9, we present a detailed visualization of prediction results derived from three distinct regressors, applied to both real-world datasets and synthetic graph datasets based on the Clustering Coefficient (Syn-CC). These visualizations provide a comprehensive comparison of the performance of each regressor, illustrating their capacity to accurately predict the metric of Effectiveness across varying data conditions.

All three regressors showcased notable proficiency in predicting Effectiveness correctly. This uniform performance across diverse regressors underscores the robustness of the Effectiveness measure in capturing pertinent graph features.

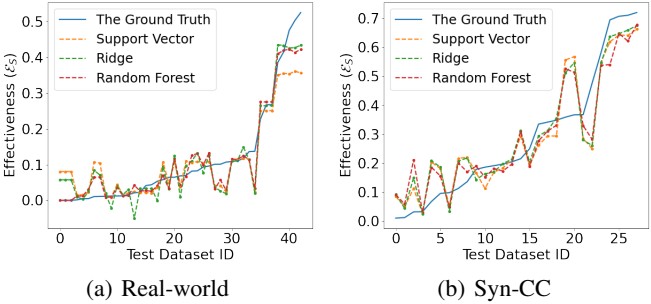

(a) Real-world           (b) Syn-CC

Figure 9: Effectiveness prediction using different regressors on different scale datasets.

### A.4 HYPERPARAMETERS IN OUR EXPERIMENTS

Table 4: Hyperparameters in our experiments.

| Models | Input Features | Layers | Learning rates | Batch sizes | Hidden Units | Optimizers | Aggregations |
|---|---|---|---|---|---|---|---|
| $\mathcal{M}_S^{\mathbf{Baseline}}$ | Avg.Degree | [2, 3] | [1e-2, 1e-3, 1e-4] | [64, 128] | 300 | Adam | - |
| $\mathcal{M}_A^{\mathbf{Baseline}}$ | Node Attributes | [2, 3] | [1e-2, 1e-3, 1e-4] | [64, 128] | 300 | Adam | - |
| $\mathcal{M}_S^{\mathbf{GCN}}$ | Node degree | [3, 4, 5] | [1e-2, 1e-3] | [64, 128] | [64, 128, 300] | Adam | [mean, sum] |
| $\mathcal{M}_A^{\mathbf{GCN}}$ | Node attributes | [3, 4, 5] | [1e-2, 1e-3] | [64, 128] | [64, 128, 300] | Adam | [mean, sum] |
| $\mathcal{M}_S^{\mathbf{GIN}}$ | Node degree | [3, 4, 5] | [1e-2, 1e-3] | [64, 128] | [64, 128, 300] | Adam | [mean, sum] |
| $\mathcal{M}_A^{\mathbf{GIN}}$ | Node attributes | [3, 4, 5] | [1e-2, 1e-3] | [64, 128] | [64, 128, 300] | Adam | [mean, sum] |

All experiments were conducted by the benchmark framework with 10-cross validation, an experiment process will search for the best hyperparameters from each configuration list in Table 4, by using the risk assessment and model selection schemes Errica et al. (2020).

A.5 PROOF OF EQUATION 2

We provide the proof that the $\mathcal{E}$ of each type in following equation varies from 0 to 1, if the worst accuracy equals to random guessing.

$$\mathcal{E}(D) = \sum_{\mathbf{type} \in \{\mathbf{S},\mathbf{A}\}} \frac{|\delta_{\mathbf{type}}(D)|}{R^*(|Y|-1)} \cdot \frac{1-R^*}{1-|Y|^{-1}}, \text{ where } R^* = \min(R1, R2). \tag{8}$$

**Proof:** We assume $R^* \in [|Y|^{-1}, 1]$, and denote the higher accuracy as $R_h \in [R^*, 1]$, the first component as $C1(R_h, R^*) = \frac{R_h - R^*}{R^*(|Y|-1)}$. If we fix $R^*$, then $C1$ is monotonically increasing as $R_h$ increases, so that $C1 \in [0, \frac{1-R^*}{R^*(|Y|-1)}]$. Obviously, $C1$ is monotonically decreasing as $R^*$ increases, so that $C1 \in [0, \frac{R_h - |Y|^{-1}}{1-|Y|^{-1}}]$. Thus $C1$ obtain maximum value when $R_h = 1, R^* = |Y|^{-1}$, i.e., $C1(1, |Y|^{-1}) = 1$. The minimum value is 0 for any $R_h = R^*$. The second component $\frac{1-R^*}{1-|Y|^{-1}}$ is obviously ranging from 0 to 1, when $R^*$ decreases from 1 to $|Y|^{-1}$, two components obtain maximum value 1 when $R_h = 1, R^* = |Y|^{-1}$. Therefore, the multiplication of two components varies from 0 to 1.

More properties of this definition can be found in Sec3.2.

A.6 ADDITIONAL ANALYSIS OF RELATION BETWEEN GRAPH PROPERTIES AND EFFECTIVENESS

We provide more analysis on the relation of graph properties and effectiveness beyond the sec 4.1 in the main paper. In Sec 4.1, we show the correlations of each main properties and label sequence. Based on the definition of effectiveness, there are two propositions for better illustration of such relation.

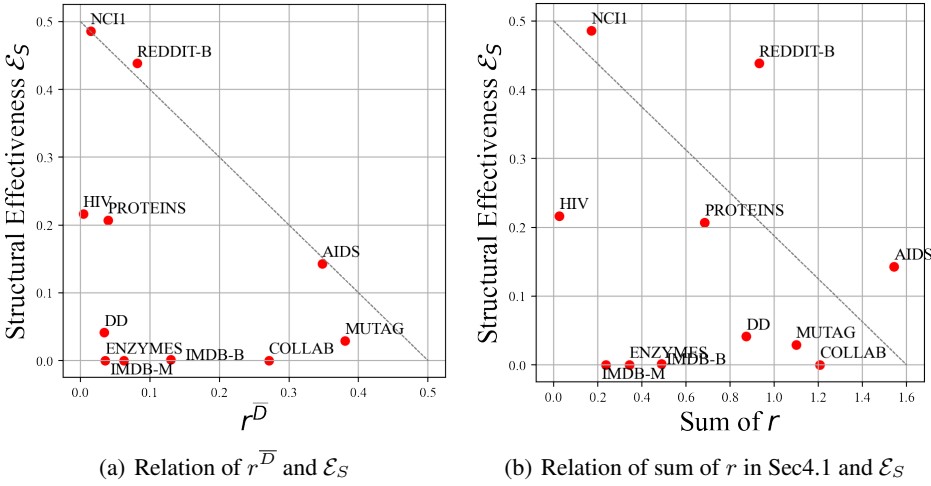

(a) Relation of $r^{\overline{D}}$ and $\mathcal{E}_S$      (b) Relation of sum of $r$ in Sec4.1 and $\mathcal{E}_S$

Figure 10: The relation of graph properties and structural effectiveness.

**Claim 1.** *If a graph property sequence is obvious correlated with label sequence, then it is easy for a baseline method to obtain a higher accuracy by using that property.*

**Proposition 1.** *If claim 1 holds, then if a GNN method can extract that property well without directly using that property, then the effectiveness must be low. In contrast, if GNN method cannot extract that property, the effectiveness must be high.*

**Claim 2.** *If a graph property sequence is almost uncorrelated to the label sequence, then a baseline method using that property leads to low accuracy.*

**Proposition 2.** *If claim 2 holds, then if a GNN method can extract that property well, the effectiveness must be high, in contrast, if GNN method cannot extract that property, the effectiveness must be low.*

Figure 10(a) demonstrates the relation between average degree correlation $r^{\overline{D}}$ and structural effectiveness $\mathcal{E}_S$, basically, higher $r^{\overline{D}}$ leads to lower $\mathcal{E}_S$. For those datasets, it is consistent with the Proposition 1 and Proposition 2. However, for the datasets such as DD, ENZYMES, IMDB-M, IMDB-B, even with low correlations, the effectiveness is still low, this may because of that the GNN method is unable to extract any useful features, leading to equivalent performance as baseline methods.

Figure 10(b) demonstrates the relation between the summation of all correlation efficients in Sec4.1 and structural effectiveness $\mathcal{E}_S$. We found that some datasets with low effectiveness such as PROTEINS, DD shift from left to right, this means other highly correlated properties instead of average degree are the reasons that lead to low effectiveness.

In the future work, it is worth to build a theoretical analysis that relates the graph property and the ability of GNNs for extracting that property.

### A.7 ATTRIBUTE VS STRUCTURE

In attributed datasets, graph learning methods need to simultaneously capture information of attributes and structures for downstream tasks. Therefore, understanding the correlation between these two types of information and their impact on downstream tasks is crucial. In this section, we partition attributed datasets into attribute-dominant and structure-dominant categories based on their respective contributions to downstream tasks.

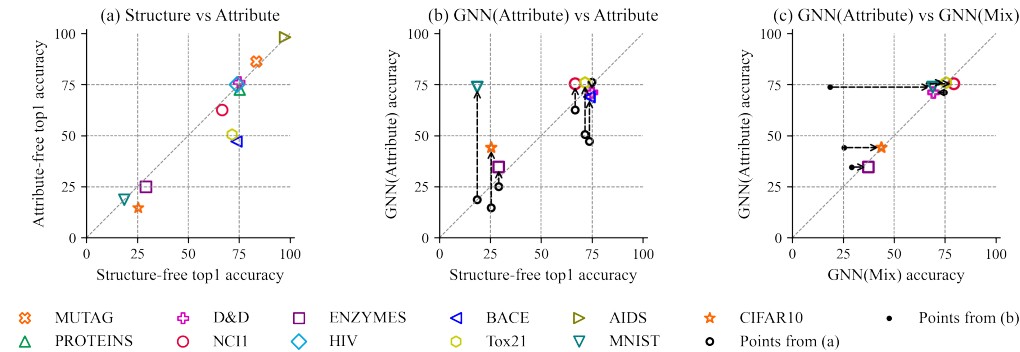

Figure 11: Performance shits via different information input.

The Figure 11 illustrates the accuracy shift in accuracy by manipulating inputs with different types of information. In Figure 11(a), *attribute-free top1 accuracy* represents the highest accuracy obtained by methods (including GIN, GCN, graph kernels, MLPs) with only taking structural information (without attributes) as input. The *structure-free top1 accuracy* represents the highest accuracy obtained by methods (MLPs, MolecularFingerprint) with only taking attribute information (without structural information). Note that, the SMILE string as input actually contains partial structural information.

In Figure 1, the points along the 45-degree axis represent similar contributions to classification under two distinct information inputs. Additionally, only a few points lie below the 45-degree axis, indicating that for a minority of datasets, node attributes play a significant role. It's important to reiterate that the node attributes used here conceal a piece of information: the lack of consideration for interactions between nodes.

Moving forward, in Figure 2, we feed node attribute information into the GNN method which enhanced inter-node interactions, resulting in values along the vertical axis, while keeping the horizontal axis consistent with Figure 1. We found that most datasets shift upper. Due to the GNN's inherent ability to capture graph structural information, these dual factors cause several points to shift upward, resulting in improved accuracy. Notably, the effect is most pronounced for CIFAR10 and MNIST. For illustration clarity, nodes with movement less than 4% have been omitted.

In Figure 3, GNN(Mix) represents a GNN approach where node attributes are extended by concatenating average degree information into the node features as input. The resulting accuracy from this approach is plotted on the horizontal axis. Almost all datasets align with the 45-degree axis, implying that additional feature enhancement is unnecessary. GNN can inherently learn graph structural information through node attributes. Importantly, this implies an intriguing observation: distinguishing whether downstream tasks are driven by attribute or structural information when GNN takes node attributes as input is challenging. This underscores the need to investigate whether a dataset is attribute-dominant or structure-dominant. This distinction is pivotal for selecting datasets to study GNN's expressive capabilities.

Some interesting findings are as follows:

- In most attributed datasets, attributed baselines (without any graph structural information) and structure-agnostic methods (including GNNs and graph-kernel methods, without any node attribute information) achieve similar levels of performance (Figure 5a).

- GNNs can improve the node interactions, more than just capture the structure of graphs. Specifically, when GNNs exclude node attributes, their performance is comparable to attribute-based baselines. Yet, incorporating node attributes notably boosts GNNs' accuracy, attributed to heightened attribute interactions rather than structural considerations (Figure 5b).

- The interaction information between nodes is more crucial than using node information alone or using structural information alone (Figure 5b).

- Additional artificial node features (random noise, average degree, etc) do not significantly enhance the performance of GNNs (Figure 5c).

### A.8 OVERALL PROTOCOL FOR EVALUATION OF EFFECTIVENESS

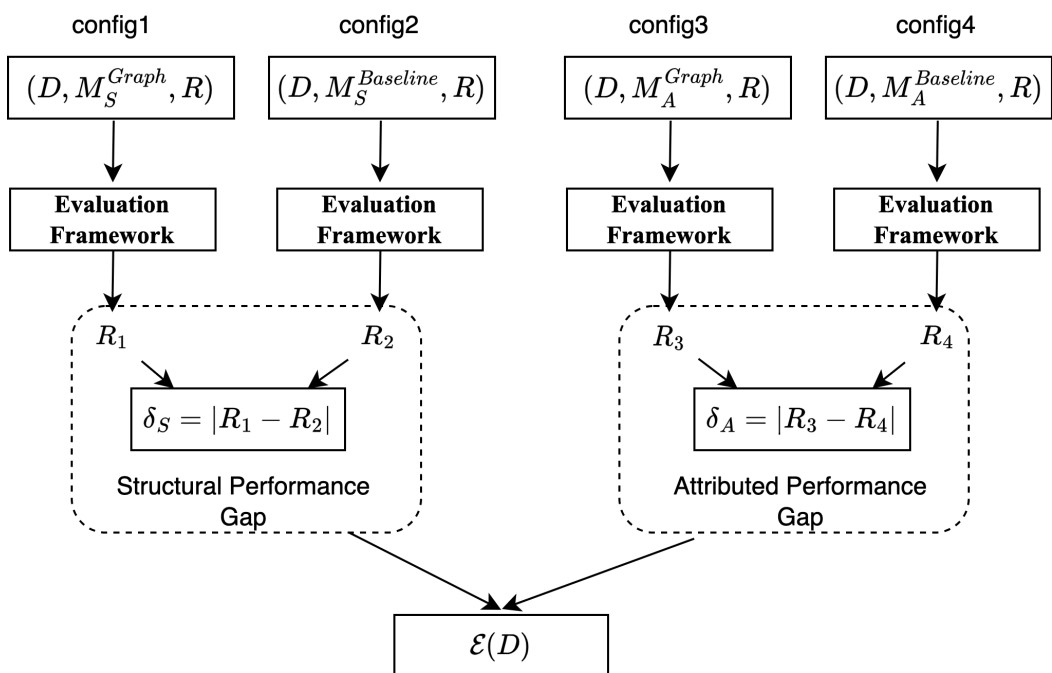

Figure 12: Overall protocol for evaluation of effectiveness. The **Evaluation Framework** is mainly based on the work Errica et al. (2020), refer to Figure 13.

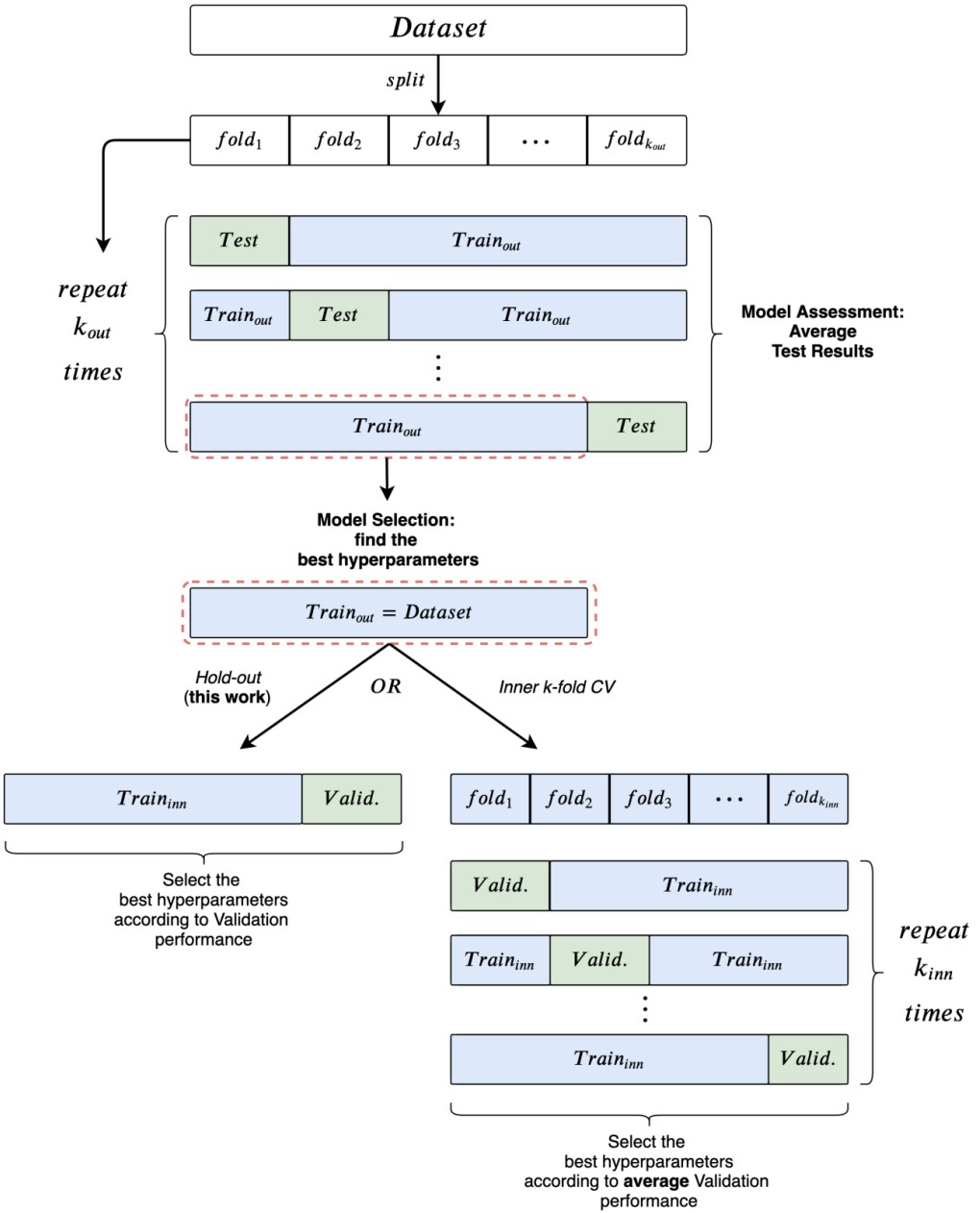

Figure 13: Visual representation of the **Evaluation Framework**. This is a reproduced figure from Errica et al. (2020). For more details, readers are encouraged to consult the original paper.

