# OpenReview forum: "Rethinking the Effectiveness of Graph Classification Datasets in Benchmarks for Assessing GNNs"
_ICLR.cc/2024/Conference — ICLR 2024 Conference Withdrawn Submission_

### Official Review · Reviewer_DPDx · 2023-10-31

**Soundness:** 3 good
**Presentation:** 2 fair
**Contribution:** 2 fair
**Rating:** 5
**Confidence:** 3

**Summary:**

The authors propose a benchmark for the graph classification task. Specifically, they curated datasets and GNN baselines, defined new evaluation metrics, and conducted evaluation experiments. Additionally, they developed a synthetic graph data generator and confirmed its ability to control the correlation between simple graph statistics and class labels.

**Strengths:**

S1. The new evaluation metric, effectiveness metric, can capture the difficulty of a dataset based on the number of classes and the absolute accuracy. The experimental results (Figure 2) demonstrate its effectiveness.

S2. The authors introduce Theorem 1 and utilize it to propose a synthetic graph generator that controls the correlation between graph statistics and class labels.

S3. The authors successfully reveal some characteristics of representative GNSSs, such as GIN, using various datasets generated by the graph generator.

**Weaknesses:**

W1. The synthetic graph generator is based on the Erdos-Renyi (ER) graph model, which may not be practical for generating many real-world graphs. For instance, in fields like chemistry and pharmacy, the motif structure is crucial, and simple metrics such as node count and cycle count are not sufficient for graph classification.

W2. While the authors conducted a correlation analysis between graph statistics and class labels produced from the graph generator, the quality analysis for the generated graphs is not enough because the accuracy evaluations of graph classification are not conducted.

W3.  The figures and tables should be improved.
- The organization of Figure 2 and Figure 3 differs (stacking or non-stacking), making it difficult to compare them easily.
- The font size in many figures is too small to read.

W4. The sources for the 14 datasets used in Section 4.3 are not provided.

**Questions:**

Q1. Can you provide some representative application fields and supporting literature where simple metrics like node count and cycle count are good enough to effectively estimate class labels with high accuracy?

Q2. The reason for limiting the cycle count sequence to k = {3, 4, 5, 6} is unclear.

---

> ### Author Response · Authors · 2023-11-14
> **Response to reviewer DPDx**
>
> We sincerely appreciate your thorough review and the insightful feedback on our manuscript. And we have modified the manuscript according to you suggesionts in the latest revison. Below, we respond to your questions.
>
> **[Weakness 1] Concern about the practicality of the Erdos-Renyi (ER) graph model in generating real-world graphs.**
>
> Our synthetic graph generator, exemplifying the application of Theorem 1, is designed to demonstrate the control of correlation between graph properties and labels. Theorem 1 is versatile and can be adapted to any graph generation process that specifies numerical graph properties, making it suitable for real-world applications.
>
> **[Weakness 2] Insufficient quality analysis of generated graphs due to a lack of accuracy evaluations in graph classification.**
>
> Sorry for the confusion. We indeed provide the accuracy evaluations and performance gaps of these generated graphs in Section 4.2, specifically in Figure 6(b). This figure illustrates the relationship between property correlation and performance gaps. Furthermore, we demonstrate performance gaps ranging from 0% to 100%, associated with correlations from 0 to 1. This figure suggests the effectiveness of our controllable graph generation method.
>
>
> **[Weakness 3] Small font size in figures making them difficult to read.**
>
> In response to this feedback, we have increased the font size in all figures in our latest revision to enhance readability.
>
> **[Weakness 4] Lack of sources for the 14 datasets used in Section 4.3.**
>
> Sorry for that, this is a typo, it should be all 16 datasets, we have corrected it in the latest revison.
>
> **[Question 1] Request for examples and literature where simple metrics like node count and cycle count effectively estimate class labels.**
>
> In our introduction, we reference studies [1,2,3] demonstrating that MLP-based baselines, which utilize metrics like average degree and clustering coefficients (cycle count=3), can perform comparably or even better than GNNs. Other research [4,5] emphasizes the importance of counting graph substructures, crucial in biology, chemistry, and social network analysis, particularly highlighting the role of cycle counts in chemical substructures. For more complex properties, such as graph biconnectivity, essential in fields like chemistry and social networks, a recent study [6] shows that popular GNN paradigms struggle with tasks efficiently handled by simpler algorithms. Node count, while basic, is implicitly included in metrics like average degree or clustering coefficients and shows high correlation with labels in datasets like REDDIT. Given that GNNs are at least as powerful as the 1-WL test, which easily distinguishes based on node count, GNNs should theoretically handle this metric effectively.
>
> **[Question 2] Rationale for limiting the cycle count sequence to k = {3, 4, 5, 6}.**
>
> This limit is informed by our finding that significant graph properties in social networks and molecular datasets, such as clustering coefficients (k=3) and common structures like benzene rings (k=6), are adequately represented within this range. Cycle counts larger than 6 are relatively rare in these datasets, hence their exclusion.
>
>
> References:
>
> [1] Federico Errica, Marco Podda, Davide Bacciu, and Alessio Micheli. "A fair comparison of graph neural networks for graph classification." In ICLR. OpenReview.net, 2020.
>
> [2] Qi Zhao and Yusu Wang. "Learning metrics for persistence-based summaries and applications for graph classification." Advances in Neural Information Processing Systems, 32, 2019.
>
> [3] Vijay Prakash Dwivedi, Chaitanya K. Joshi, Anh Tuan Luu, Thomas Laurent, Yoshua Bengio, and Xavier Bresson. "Benchmarking graph neural networks." Journal of Machine Learning Research, 24 (43):1–48, 2023
>
> [4] Huang, Yinan, Xingang Peng, Jianzhu Ma, and Muhan Zhang. "Boosting the Cycle Counting Power of Graph Neural Networks with I$^2$-GNNs." arXiv preprint arXiv:2210.13978 (2022).
>
> [5] Chen, Zhengdao, Lei Chen, Soledad Villar, and Joan Bruna. "Can graph neural networks count substructures?." Advances in neural information processing systems 33 (2020): 10383-10395.
>
> [6]Zhang, Bohang, Shengjie Luo, Liwei Wang, and Di He. "Rethinking the expressive power of gnns via graph biconnectivity."  In ICLR. OpenReview.net, 2023.

---

> > ### Comment · Reviewer_DPDx · 2023-11-23
> >
> > Thanks for the detailed responses.
> > >>W1. The synthetic graph generator is based on the Erdos-Renyi (ER) graph model, which may not be practical for generating many real-world graphs. For instance, in fields like chemistry and pharmacy, the motif structure is crucial, and simple metrics such as node count and cycle count are not sufficient for graph classification.
> >
> > > [Weakness 1] Concern about the practicality of the Erdos-Renyi (ER) graph model in generating real-world graphs.
> > Our synthetic graph generator, exemplifying the application of Theorem 1, is designed to demonstrate the control of correlation between graph properties and labels. Theorem 1 is versatile and can be adapted to any graph generation process that specifies numerical graph properties, making it suitable for real-world applications.
> >
> > The simple graph properties listed in Section 4.1 do not seem good enough to capture the graph features. We also need to distinguish other typical graph features, either random graph/scale-free graph or homophily/heterophily/non-homophilous graphs.  This reply also relates to Question 1.
> >
> > > [Weakness 2] Insufficient quality analysis of generated graphs due to a lack of accuracy evaluations in graph classification.
> > Sorry for the confusion. We indeed provide the accuracy evaluations and performance gaps of these generated graphs in Section 4.2, specifically in Figure 6(b). This figure illustrates the relationship between property correlation and performance gaps. Furthermore, we demonstrate performance gaps ranging from 0% to 100%, associated with correlations from 0 to 1. This figure suggests the effectiveness of our controllable graph generation method.
> >
> > I agree that Figure 6(b) is a useful accuracy evaluation.

---

### Official Review · Reviewer_EXhn · 2023-10-31

**Soundness:** 3 good
**Presentation:** 2 fair
**Contribution:** 3 good
**Rating:** 8
**Confidence:** 4

**Summary:**

In the context of graph classification, the paper presents:
1) an empirical protocol to benchmark GNN against structure-based and attribute-based simple baselines;
2) a novel metric to quantify the effectiveness of graph datasets, i.e. to establish if the dataset is suitable to evaluate novel methods;
3) an algorithm to generate synthetic datasets with controlled correlation between graph properties and graph labels.

**Strengths:**

- the work nicely extends previous work in the field. A measure to quantify dataset effectiveness is timely in a field where often progress is measured on flawed benchmarks.
- sufficient novelty: a novel metric and a method to generate controlled datasets are proposed, which to my knowledge are novel contributions.
- thorough experimental assessment, both in width and depth.

**Weaknesses:**

Not much to say, this paper is well written and the contribution is very welcome in the field. If I have to be picky, I'd say:
- lack of recommendations to the practitioner (see below)

Minor:
- there are many typos, which I suspect come from a rushed writing. Please have a second round of proofread before the rebuttal.

**Questions:**

- Do you have recommendations for GNN practitioners? e.g., which benchmarks should definitely be avoided when presenting a novel graph-based method?

---

> ### Author Response · Authors · 2023-11-14
> **Response to reviewer EXhn**
>
> We sincerely thank you for your thoughtful appreciation of our work. In the latest revision, we have addressed and corrected all typos as per your suggestions.
>
>
> **[Question 1] Do you have recommendations for GNN practitioners?**
>
> Thank you for highlighting this important aspect! We have offered following disccussion in the latest revison to address your question:
>
> While the choice of datasets should not solely depend on the effectiveness metric as proposed in our paper, this metric serves as a valuable indicator to understand the characteristics of these datasets. For instance, our methodology focuses on comparing a limited set of graph properties as baselines against GNN performance. However, there are additional graph properties that are crucial for accurate predictions.
>
> Our study is to bridge the gap between theoretical understanding and practical application in the realm of graph classification datasets and is to provide a comprehensive empirical analysis that could inform and guide future theoretical explorations in this field.
>
> As we have summarized in Section 5, exploring a broader range of properties and establishing a relationship between GNN expressiveness and these properties is essential to comprehending what GNNs truly learn from datasets.
>
> Therefore, for practitioners, our findings provide valuable insights. Specifically, we highlight that datasets with a high correlation between average degrees and labels may be too simplistic for GNNs, making it challenging to discern the additional learning and predictive capabilities of these networks.
>
> In practical terms, this implies that when selecting datasets for training and evaluating GNNs, practitioners should consider the complexity and diversity of graph properties present. Datasets that offer a wider range of challenges will likely be more effective in distinguishing the unique strengths of GNN methodologies.

---

> > ### Comment · Reviewer_EXhn · 2023-11-22
> > **Thank you**
> >
> > I have acknowledged you response, and I'm leaving my score unchanged. Good luck!

---

### Official Review · Reviewer_eLa4 · 2023-11-01

**Soundness:** 3 good
**Presentation:** 3 good
**Contribution:** 3 good
**Rating:** 6
**Confidence:** 3

**Summary:**

This paper investigates the important problem in graph classification, which is why certain GNN methods cannot outperform even simple baselines such as MLPs. The authors conduct extensive experiments to showcase the reasons for such a situation, and further propose a fair measurement that considers the performance gap and the inherent complexity of the datasets. The authors further propose a novel algorithm that can generate controllable correlation datasets. The experiments are comprehensive and showcase meaningful observations. The authors also provide theoretical analysis to support their claim

**Strengths:**

1. This paper investigates the important problem in graph classification, which is why certain GNNs cannot outperform simple baselines such as MLPs. The authors conduct extensive experiments on 16 datasets to quantify the performance gap between different methods and using only structural and attribute information.

2. The authors propose a novel measurement that can comprehensively assess the performance advancement of GNNs on graph classification tasks, while considering the effects of both the changing performance portion and the complexity of datasets.

3. The authors propose a novel algorithm that can generate controllable correlation datasets for evaluation.

**Weaknesses:**

1. There are several grammatical errors in the paper. For example, "whether a GNN method has truly improved" should be "whether a GNN method has been truly improved".

2. The authors do not provide a comprehensive figure to illustrate the overall assessment of the proposed strategy for graph classification methods. Although the idea is straightforward, such a figure can help readers capture the main objective for understanding.

3. The effectiveness definition in Eq. (2) seems not to be intuitive. The authors multiple two factors that consider both the changing proportion of the performance gap and the complexity. However, it remains unclear why we should use the product form instead of others. Also, it is unclear why the first component is normalized by |Y|-1 instead of |Y|.

**Questions:**

Have the authors considered devising a method that can solve the limitations in existing graph classification methods?

---

> ### Author Response · Authors · 2023-11-14
> **Response to reviewer eLa4**
>
> We are grateful for your constructive suggestions and insightful feedback. In our latest manuscript revision, we have diligently corrected all grammatical errors and made adjustments based on your valuable comments. These changes include enlarging the font size in figures for better clarity.
>
> Below, we respond to your concerns.
>
> **[Weaknesses 1]There are several grammatical errors in the paper.**
>
>  We have thoroughly reviewed the entire paper to correct these grammatical errors in the latest revison.
>
> **[Weaknesses 2]The authors do not provide a comprehensive figure to illustrate the overall assessment of the proposed strategy for graph classification methods.**
>
> Thank you for pointing out this important aspect. In response to your valuable feedback, we have included a comprehensive figure in Appendix A8 of our latest revision. This figure aims to clarify our overall framework, enhancing the reader's understanding and facilitating the replication of our results. We appreciate your suggestion and hope that this addition effectively addresses your concern.
>
> **[Weakness 3] Concerning the definition of effectiveness in Eq. (2), the use of the product form and the rationale behind normalizing the first component by \|Y|-1 instead of \|Y| may seem complex.**
>
> We apologize for any lack of clarity in our explanation and appreciate the opportunity to elaborate further. Here, we try to provide the intuitions, more detailed explanation and proof are provided in Appendix 5, with key properties discussed in Sec. 3.2.
>
> Briefly:
>
>    1. We consider the worst-case scenario where accuracy equals random guessing, which is 1/|Y|, e.g., if |Y| = 4, then random guessing has accuracy 25%.
>    2. The largest gap in performance is therefore 100% - 1/|Y|.
>    3. The proportion $(R_{high} - R_{low})/ R_{low} \in [0, (1 - |Y|^{-1})/|Y|^{-1}] = [0, |Y| - 1]$.
>    4. Thus, we normalize the proportion by $|Y| - 1$, such that the proportion ranges from 0 to 1.
>
>
> **[Question 1] Regarding devising methods to address existing limitations in graph classification**
>
> Our primary focus is on the effectiveness of benchmark datasets rather than directly improving GNN methods. That said, our findings can assist in assessing the fairness of new methods. Notable recent studies [1-5] have made significant strides in capturing complex structures, a crucial area where basic GCN and GIN models fall short. These works often employ synthetic datasets to test method effectiveness, underscoring the community's need for more robust real-world datasets.
>
>
> References:
>
> [1] You, Jiaxuan, Jonathan M. Gomes-Selman, Rex Ying, and Jure Leskovec. "Identity-aware graph neural networks." In Proceedings of the AAAI conference on artificial intelligence, vol. 35, no. 12, pp. 10737-10745. 2021.
>
> [2] You, Jiaxuan, Rex Ying, and Jure Leskovec. "Position-aware graph neural networks." In International conference on machine learning, pp. 7134-7143. PMLR, 2019.
>
> [3] Huang, Yinan, Xingang Peng, Jianzhu Ma, and Muhan Zhang. "Boosting the Cycle Counting Power of Graph Neural Networks with I$^2$-GNNs." arXiv preprint arXiv:2210.13978 (2022).
>
> [4] Chen, Zhengdao, Lei Chen, Soledad Villar, and Joan Bruna. "Can graph neural networks count substructures?." Advances in neural information processing systems 33 (2020): 10383-10395.
>
> [5] Zhang, Bohang, Shengjie Luo, Liwei Wang, and Di He. "Rethinking the expressive power of gnns via graph biconnectivity."  In ICLR. OpenReview.net, 2023.

---

> > ### Comment · Reviewer_eLa4 · 2023-11-23
> >
> > Thanks for your response.

---

### Official Review · Reviewer_yi6y · 2023-11-02

**Soundness:** 2 fair
**Presentation:** 3 good
**Contribution:** 2 fair
**Rating:** 5
**Confidence:** 3

**Summary:**

This paper revisits some datasets and claims that some graph benchmarks are unable to distinguish the advancements of GNNs over other methodologies. In particular, the authors propose an empirical protocol for evaluating the dataset discriminability. In this protocol, the authors use the absolute error between Graph-based methods and other methodologies to quantify their performance gap from two different perspectives (structure and attribute). Then they use this protocol to revisit and analyze some existing datasets.

The paper then points out the limitations of their protocol and designs a metric that quantifies the effectiveness of a dataset. Specifically, the proposed metric alsk takes the number of classes, and the worst model performance into consideration.

Furthermore, the authors investigate the reasons behind the low effectiveness of datasets. Specifically, they investigate the correlations between various properties and sample labels. Base on the understanding, they propose a method for generating controllable graphs.

**Strengths:**

*Clarity*. The research questions and goals of this paper are stated clearly, and the whole paper is written in a clear logic, analyzing and solving the problem step by step. For example, the authors find some limitations for the protocol in section 2, so they propose a new metric in section 3.

*Quality*. More empirical evidence is provided in the appendix to illustrate the effectiveness of the proposed metric. The details of the experiments like the hyper-parameters for training are also provided in the appendix which will facilitate other researchers to reproduce the results.

*Significance*. This paper revisits some existing datasets or benchmarks and reminds people that some datasets are unable to distinguish the advances of Graph-based models. This problem is very important. Solving this problem holds substantial impact potential for the broader community.

**Weaknesses:**

1. In general, the novelty of the paper is a bit limited. The paper mainly concentrates on an empirical investigation of the effectiveness of existing benchmarking datasets for graph classification. While providing some insights on the datasets, there is a lack of theoretical understanding or support. This also makes it more suitable for a benchmark track such as NeuIPS benchmarking track.

2. Some of the choices and designs in the paper are not well-motivated.
a.  For example, why GCN and GIN are chosen as the two methods for evaluating the benchmarks. The authors claim “GIN is spatial” and “GCN is spectral”, which is not informative enough. It would be better if the authors could provide more description and comparison between the two (or with other methods) and motivate their choices.
b. Furthermore, the choice of average degree as the input for the structure-dominated MLP model seems to be a bit ad-hoc without solid support. It is not very clear why the average degree is selected especially given that the authors investigate several other graph properties in later sections.

3. It is not very clear how Theorem 1 describes or supports the graph generation process. In particular, the meaning of Eq. (3) is not clearly explained. It would be helpful if the authors could provide more details on this part. Furthermore, as generating synthesized graphs is a key component of the paper, it might be better if the authors provide more details on the graph generation model and process.

**Questions:**

Please answer the questions raised in the section on weakness. In addition, there are a few other questions.

1. In Observation 3, of Section 2.3, it claims “only REDDIT-B displayed a noteworthy performance gap, indicating a strong correlation between degree information and task labels”. Why this is the case, if there is a strong relation between average degree and task labels, should the performance gap be small as the baseline can also achieve good performance?

2. In section 4.1, how is the correlation between graph properties and class labels calculated? Can you provide any details in calculating this correlation? Also, how high should these correlation coefficients be to be considered high correlations?

---

> ### Author Response · Authors · 2023-11-14
> **Response to reviewer yi6y (Part I)**
>
> We thank the reviewer for their insightful comments and constructive feedback. We have modified the latest revison according to your suggestions.
>
> We are particularly appreciative of your recognition of the paper's clarity, the comprehensive empirical evidence provided, and its potential significance in the field of graph neural networks (GNNs).
>
> Below, we respond to your concerns.
>
> **[weakness 1] In general, the novelty of the paper is a bit limited**
>
> We first thank you for acknowledging our contributions in your comments: "Significance. This paper revisits some existing datasets or benchmarks and highlights the issue that some datasets fail to distinguish advancements in Graph-based models. This problem is critically important. Addressing it has the potential to significantly impact the broader community."
>
> To the best of our knowledge, **our work is the first to thoroughly study and provide an explicit definition for the effectiveness of datasets in the graph learning area, which is an urgent issue that requires investigation.** Previous excellent works [1,2,3,4,5,6] have observed that the expressiveness of GNN is not fully exploited due to issues with dataset reliability, where even baseline models can achieve comparable performance.
>
> Therefore, we have meticulously studied this problem by introducing a novel definition of effectiveness. Moreover, Theorem 1 provides a powerful tool for controllable graph generation, facilitating future explorations in creating controllable synthetic datasets. Our extensive results on 16 common benchmarks in the field and synthetic datasets validate the effectiveness of this definition. Looking ahead, we believe our work can be further refined for greater generality and accuracy.
>
> In addition, we acknowledge your concern about the perceived lack of theoretical support. However, it is important to emphasize that **the primary objective of our study is to bridge the gap between theoretical understanding and practical application in the realm of graph classification datasets. Our approach is to provide a comprehensive empirical analysis that could inform and guide future theoretical explorations in this field.**
>
> Moreover, our work aligns well with the scope of ICLR, under the major area: datasets and benchmarks. Our research contributes to this by offering a detailed empirical analysis, highlighting the strengths and limitations of current benchmarking datasets, thereby setting the stage for more theoretical investigations.
>
> We believe that our study, with its focus on empirical analysis, complements theoretical research and provides a necessary foundation for developing more robust and effective graph classification methods. The insights gained from our work can spur further theoretical research, contributing to a more comprehensive understanding of graph classification benchmarks.
>
>
> **[weakness 2.1] why GCN and GIN are chosen as the two methods for evaluating the benchmarks?**
>
> As detailed in Section 2.1, we selected GIN and GCN for our study because these two Graph Neural Networks (GNNs) represent the most fundamental types in both spatial and spectral categories. They are widely used and demonstrate competitive performance in most applications [7,8]. These GNNs are also considered baseline models on the Open Graph Benchmark platform (OGB) [3]. Furthermore, GIN [9] is proven to be as powerful as the 1-WL test and is a quintessential example of message-passing neural networks. Meanwhile, GCN [10] is known for achieving competitive performance on small graphs with just 2 layers.
>
>
>
> **[weakness 2.2] Furthermore, the choice of average degree as the input for the structure-dominated MLP model seems to be a bit ad-hoc without solid support.**
>
> Following the methodologies of previous works [1,2,3,4,5,6], we utilize degree information, without node attributes, as the input for MLPs, or employ the aggregation of neighboring features [3], which also implicitly includes degree information. Given that the average degree is a significant graph property in various applications, and considering that the learning paradigm of GNNs involves aggregating neighboring information followed by a pooling layer, GNNs can readily learn node degree information. Thus, the critical aspect of our study is to determine whether GNNs learn solely the average degree information or if they capture more complex structural information as well.

---

> ### Author Response · Authors · 2023-11-14
> **Response to reviewer yi6y (Part II)**
>
> **[weakness 3] It is not very clear how Theorem 1 describes or supports the graph generation process.**
>
> Apologies for any confusion caused. Theorem 1 plays a crucial role in controllable graph generation, specifically with given correlation coefficients, as detailed in Appendix Algorithm 1. We hope the following explanation will help address your concerns:
>
> Utilizing Theorem 1, we are able to efficiently generate the label sequence for the graph generation process in the following manner:
>
>
> ### Algorithm 1: Controllable dataset construction
> **Input**: {$r_k$}$_{k=1}^{K}$, number of labels: $C$,
>
>  {$P_k$}$_{k=1}^K \sim N(\mu_k, \sigma_k)$ or $U(a_k, b_k)$;
>
> **Output**: Dataset $D$ with size $N$;
>
> **for** k=1 to K do;
>
> **Sample** $n_k \sim N(0, \sigma_k)$ or  $U(-\sqrt{3}, \sqrt{3})$ with size $N$;
>
> $P_k = \mu_k + \sigma_k n_k$ or $\frac{a_k + b_k}{2} + \sqrt{\frac{b_k - a_k}{12}} n_k$;
>
> **end**
>
> **Calculate** $Y$ by the Equation (3);
>
> $Y = \text{ROUND}(\text{NORM}(\mathcal{Y}) * C)$;
>
> D = {(g_i, y_i)}$_{i=1}^N$,
>
> where each graph $g_i$ has properties {$P_k[i]$}$_{k=1}^K$,  and corresponding label $y_i = Y[i]$;
>
>
>
>
> **[Question 1] In Observation 3, of Section 2.3,why this is the case?**
>
> Thank you for pointing out that issue. Indeed, there was a typographical error; it should state "weak correlation" instead of "strong correlation." In Section 4.3 and Figure 4, we illustrate the correlations between graph properties and labels. Specifically, we observed that REDDIT-B exhibits a lower correlation between the degree of nodes and labels compared to other properties. This suggests that using average degree as a baseline input is less effective than employing GNNs. In contrast, other social network datasets such as COLLAB and REDDIT-M show higher correlations, leading to better performance.
>
> **[Question 2.1] Can you provide any details in calculating this correlation?**
>
> Thank you for your query regarding the calculation of the correlation. We have attempted to explain this in Section 4.2 and Theorem 1, where we mention the use of Pearson correlation for calculating the coefficients. We understand that these sections might not have been as clear as desired. To offer further clarity, we have provided a detailed step-by-step process in Algorithm 1, located in Appendix A2. This algorithm is designed to assist in applying Theorem 1 for generating graphs with specified correlation coefficients.
>
> **[Question 2.2] Also, how high should these correlation coefficients be to be considered high correlations?**
>
> You raise an important point. Indeed, the absolute values of correlation coefficients differ across various datasets and graph properties. When we refer to 'high' or 'low' correlation, it is meant for comparative purposes within similar types of graph datasets, like those in social networks. The relationship between correlation coefficients and performance gaps can be verified using synthetic datasets, as demonstrated in Figure 6(b). In the context of real-world datasets, the correlation coefficient should be regarded as one of several relative indicators.
>
>
> References:
>
> [1] Federico Errica, Marco Podda, Davide Bacciu, and Alessio Micheli. "A fair comparison of graph neural networks for graph classification." In ICLR. OpenReview.net, 2020.
>
> [2] Vijay Prakash Dwivedi, Chaitanya K. Joshi, Anh Tuan Luu, Thomas Laurent, Yoshua Bengio, and Xavier Bresson. "Benchmarking graph neural networks." Journal of Machine Learning Research, 24 (43):1–48, 2023
>
> [3] Weihua Hu, Matthias Fey, Marinka Zitnik, Yuxiao Dong, Hongyu Ren, Bowen Liu, Michele Catasta, and Jure Leskovec. "Open graph benchmark: Datasets for machine learning on graphs." Advances in neural information processing systems, 33:22118–22133, 2020
>
> [4] Will Hamilton, Zhitao Ying, and Jure Leskovec. "Inductive Representation Learning on Large Graphs." In NeurIPS, 2017.
>
> [5] Qi Zhao and Yusu Wang. Learning metrics for persistence-based summaries and applications for graph classification. Advances in Neural Information Processing Systems, 32, 2019.
>
> [6] Christopher Morris, Nils M. Kriege, Franka Bause, Kristian Kersting, Petra Mutzel, Marion Neumann. "TUDataset: A collection of benchmark datasets for learning with graphs." CoRR abs/2007.08663 (2020)
>
> [7] Zhou, Jie, Ganqu Cui, Shengding Hu, Zhengyan Zhang, Cheng Yang, Zhiyuan Liu, Lifeng Wang, Changcheng Li, and Maosong Sun. "Graph neural networks: A review of methods and applications." AI open 1 (2020): 57-81.
>
> [8] Wu, Zonghan, Shirui Pan, Fengwen Chen, Guodong Long, Chengqi Zhang, and S. Yu Philip. "A comprehensive survey on graph neural networks." IEEE transactions on neural networks and learning systems 32, no. 1 (2020): 4-24.
>
> [9] Keyulu Xu, Weihua Hu, Jure Leskovec, Stefanie Jegelka. "How Powerful are Graph Neural Networks?", ICLR 2019.
>
> [10] Thomas N. Kipf, Max Welling. "Semi-Supervised Classification with Graph Convolutional Networks.", ICLR (Poster) 2017.

---

> ### Author Response · Authors · 2023-11-18
> **Summary and additional clarifications for reviewer yi6y**
>
> We appreciate the review again. For your convenience, we have summarized 3 long responses below and included additional clarifications:
>
>
> **Novelty**:
>
>  We are the first to extensively investigate how to quantify the degree to which a dataset can serve as a benchmark in graph classification, an urgent issue in this field,  bridging the gap between theoretical understanding and practical application. We propose an insightful metric called 'effectiveness', which is easy to use for dataset selection in benchmarking.
>
> **Choice of GIN (Spatial) and GCN (Spectral)**:
>
> We chose GIN for its spatial approach, focus on the physical or 'spatial' layout of the graph. The spatial approaches such as GIN work directly with the graph's structure, treating the graph as a collection of nodes and edges, emphasizing local structure and node-level relationships in the graph. While GCN for its spectral method approximation, transform node features into the spectral domain (using the graph Fourier transform), performing operations in this domain (like filtering or convolutions), and then transforming the results back to the spatial domain. This approach is inherently global, capturing overall graph structure and relationships. This selection provides a balanced evaluation against both local and global graph features, offering a comprehensive benchmark comparison. Other GNN variants, while valuable, may not distinctly represent these two essential aspects of graph analysis as effectively as GIN and GCN for our study's scope.
>
>
>
> **how high should these correlation coefficients be to be considered high correlations?**
>
> We can set a threshold (above which is considered high) based on the effectiveness value, as there is a linear relationship between the correlation coefficients and the effectiveness value in synthetic datasets (shown in Figure 6(b)). If the correlation coefficient equals 1, then the effectiveness is 0. Generally, a higher correlation results in lower effectiveness, but the converse is not necessarily true.
>
> To further validate this, we can generate a substantial number of controllable synthetic datasets using our provided tool and calculate the correlation coefficients and effectiveness of each dataset. We then select all datasets with an effectiveness value below a predefined threshold, based on observations from real-world datasets. Next, we can establish a confidence interval for one of the graph properties of these datasets using statistical methods. A dataset with a correlation coefficient above the upper limit of the established confidence interval can be considered to have a high correlation coefficient.

---

> > ### Comment · Reviewer_yi6y · 2023-11-23
> > **Response to Authors' Response**
> >
> > Thank you for the detailed response. Many of the concerns have been addressed by the response. I will raise the score to 5.

---

### Meta-Review · Area_Chair_jzq5 · 2023-12-15

**Metareview:**

This paper investigates an important problem of GNN evaluation. Some benchmarks in this space have been long known to be unreliable, and improvements on the evaluation could have significant impact. While the paper proposes a plausible method, i.e., model performance gap, to measure the goodness of the datasets, reviewers have concerns regarding the specific design choices. Furthermore, the author responses did not convincingly address some of the concerns. For example, for the concern about the model choice, the authors responded that "GCN and GIN ... represent the most fundamental types in both spatial and spectral categories", which doesn't explain much.

As the evaluation problem is crucial to the field, the soundness of the evaluation method is even more important. Given the concerns about the proposed evaluation method, I think it is not ready for publication at its current status. The authors are encouraged to carefully leverage the reviewer comments for future submission.

**Justification For Why Not Higher Score:**

Some design choices of the proposed evaluation method are not sufficiently justified.

**Justification For Why Not Lower Score:**

N/A

---

### Decision · Program_Chairs · 2024-01-16

Reject